**Data Availability Statement:** All relevant data are within the paper and its Supporting Information files.

# Are religious patients less afraid of surgery? A cross-sectional study on the relationship between dimensions of religiousness and surgical fear

Andrija Karačić[1]*, Jure Brkić[1], Maurice Theunissen[2,3], Slavica Sović[4], Mansoureh Karimollahi[5], Branko Bakula[1], Jelena Karačić[6], David H. Rosmarin[7,8]

1 Department of General Surgery, University Hospital Sveti Duh, Zagreb, Croatia, 2 Department of Anesthesiology and Pain Management, Maastricht University Medical Center+, Maastricht, The Netherlands, 3 Department of Clinical Psychological Science, Maastricht University, Maastricht, The Netherlands, 4 Statistics Department, School of Medicine, University of Zagreb, Zagreb, Croatia, 5 School of Nursing, Ardabil University of Medical Sciences, Ardabil, Iran, 6 Department of Periodontology, Endodontology and Cariology, University Center for Dental Medicine, Basel, Switzerland, 7 Spirituality & Mental Health Program, McLean Hospital, Belmont, MA, United States of America, 8 Department of Psychiatry, Harvard Medical School, Boston, MA, United States of America

* andrija@ccm.hr

## Abstract

### Introduction

Surgical fear is common and has a negative impact on surgery and its outcome. Recent research has identified individual religiousness as an important factor among patients with associations to mental health, particularly anxiety.

### Objective

This study aimed to examine associations between religiousness and surgical fear in a representative sample of adult surgical patients in Croatia.

### Design

Cross-sectional study among elective surgery patients at different departments of a single hospital.

### Setting

University Hospital Sveti Duh, a tertiary health care facility in Croatia.

### Measurements

Religiousness and surgical fear were the variables of interest and assessed through self-report instruments. The Croatian version of the Duke Religiosity Index questionnaire (DUREL) assessed organizational religious activity (ORA), non-organizational religious activity (NORA), and intrinsic religiousness (IR). The Croatian version of the Surgical Fear

**Funding:** The authors received no specific funding for this work.

**Competing interests:** The authors have declared that no competing interests exist.

Questionnaire (SFQ) measured surgical fear and its subscales the fear of the short-term and long-term consequences of surgery. Additionally, sociodemographic characteristics and medical history were assessed. Analyses were carried out using descriptive and linear regression analyses.

## Results

178 subjects were included for data analysis. Univariate linear regression found two dimensions of religiousness (non-organizational religious activity, intrinsic religiousness) to be weak, but significant predictors of greater surgical fear (adj. $R^2$ = 0.02 and 0.03 respectively). In the multiple linear regression model together with age, gender, education and type of surgery, all three dimensions of religiousness were found to be significant independent predictors of greater surgical fear.

## Limitations

The study was single-center and cross-sectional and did not assess patients' specific religious identity.

## Conclusions

This study demonstrated significant positive associations between dimensions of religiousness and surgical fear, potentially suggesting that surgical patients experience increased religiousness to cope with heightened anxiety. Assessment and intervention to address patient religiousness is warranted in the context of surgical fear among religious patients, and the importance of religiousness in the context of surgical fear needs to be further addressed in research.

## Introduction

Surgical fear, the emotion of fear patients experience in the preoperative setting, is a common phenomenon [1], with a reported prevalence from around 11% to even 80% among adult patients [2]. It is generally accepted that surgery and various dimensions of surgery may have a negative impact on a patient´s emotional state [3]. This could be fear of the procedure itself, fear of the disease or disablement, fear of the anesthesia, or fear of the postoperative pain [4, 5]. Regardless of its cause, surgical fear has been proven to have a negative impact on surgical treatment and its outcome [6]. Since surgical fear is associated with higher levels of acute and chronic postoperative pain, it leads to increased anesthesia and analgesia administration [7]. Furthermore delayed wound healing, lesser patient adherence to the treatment plan, prolonged hospital stays and treatments, reduced physical functioning, aggravated mental health, and limited quality of life [1] all lead to increased morbidity and mortality [8] due to surgical fear. Hence surgical fear represents not only a major burden for the patient [9] but the whole health-care system [10] as well. By acknowledging the importance of surgical fear, it becomes clear that surgical fear needs to be prevented and, if possible, reduced through enhanced perioperative care and specific interventions [11–13]. It is essential for medical professionals to identify patients at risk of surgical fear. Besides factors related to surgery, such as type of surgery performed, waiting time, previous experiences with surgery and anesthesia, several

patient-related factors affect surgical fear too [14]. Examples of patient-related factors are age, sex and educational status [1, 15–17].

Recent research has focused more and more on the medical implications of a specific personal trait: religiousness [18–20]. Religiousness is a complex term and comprises one´s involvement in religious activities, attitudes and beliefs [21]. In medical literature, religiousness has been mostly studied in the context of mental health [22, 23], especially anxiety [24].

Since religiousness is a multifaceted construct [25] the relationship between religiousness and anxiety needs to be dissected into the different relationships between the specific dimensions of religiousness and anxiety, taking into account the different potential effects [26]. On the one hand, religious activity can have anxiolytic effects by activating positive cognitive schemas, enhancing emotion regulation and may act as a coping mechanism [27]. But on the other hand, because of these effects, anxiety can lead to increased religious activity, thus revealing positive associations with negative emotions [28], for example death anxiety [29]. As such, in cross-sectional studies, both positive and negative effects of religion on anxiety are often apparent; in aggregate these tend to even out each other, leading to small or null associations.

In literature there has been a surprisingly scant number of studies to explore relationships between religiousness and surgical fear or preoperative anxiety. The only study in literature whose primary intention was to investigate this relationship was by Kalkhoran and Karimollahi [2]. The researchers found an inverse relationship between religiousness and the grade of preoperative anxiety, but without statistical significance [2]. This finding was in accordance with findings from other studies on preoperative anxiety which identified religiousness to be a protective factor for preoperative anxiety [30–32]. Hence, current limited evidence suggests that patients with higher levels of religiousness would report lower levels of surgical fear.

Considering the importance of surgical fear in modern perioperative care and the need for detection of propensities for surgical fear based on patient-related factors, there is a significant gap in literature on the association of religiousness and surgical fear. We decided to conduct a study based on the work of Kalkhoran and Karimollahi [2]. The authors recommended the following for conducting the study: to conduct the study in a different religious community, in a different country and with a larger study sample while utilizing a standardized valid and reliable questionnaire for the assessment of religiousness [2]. These recommendations were implemented into our study protocol. Another justification for our study was the fact that Croatia is a predominantly religious country. The aim of our study was to investigate the association between religiousness and surgical fear and show whether religiousness was indeed a protective factor leading to lower levels of surgical fear. A secondary aim was to assess need of patients for sacral object usage in the preoperative setting.

## Materials and methods

### Study design

The study design was based on an earlier study by Kalkhoran and Karimollahi [2]. and designed as a cross-sectional study evaluating the association between two factors: religiousness and surgical fear, in surgical patients the day before surgery.

STROBE guidelines were used for reporting [33].

### Setting

The study was conducted between July 1st and September 30th 2021 on patients hospitalized at wards of the surgery department of a tertiary health care facility in Zagreb, Croatia. After hospital admission, the researchers included all patients following the criteria mentioned below. Data collection was performed on weekdays, in the afternoon before procedures, usually

between 3 p.m. and 5 p.m. The patients were informed about the study by a researcher at their bedside. After obtaining written consent, the booklet containing the self-report instruments was handed out. The patients were given time to complete the booklet at their own pace in their beds and were later collected by the nursing staff during their evening rounds.

## Participants

All patients admitted for elective surgical procedures regardless of type of surgery were candidates for study enrollment and included according to the following criteria. Inclusion criteria were consciousness, age above 18, literacy, informed consent, and being scheduled for elective surgery within a day from baseline data collection. Exclusion criteria were no proficiency in the Croatian language, illiteracy, presenting an unstable medical condition e.g. severe pain, dyspnea, severe mental disorder or cognitive impairment (e.g. dementia) hindering the patient to complete the data collection, physical disablement that hindered the patient to complete the data collection (e.g. blindness, neuromotor issues) and participation in another trial. Patients whose booklets had more than two incomplete items in either of the self-report instruments and/or incomplete surgical fear or religiousness questionnaires were excluded from the study.

## Variables

Variables of interest were religiousness and surgical fear. The levels of religiousness and surgical fear respectively were assessed through valid and reliable self-report instruments in the form of questionnaires and expressed as numerical values. Both questionnaires are dividable into subscales. Hence besides surgical fear also the fears of the short-term and long-term consequences of surgery were evaluated as well as organizational and non-organizational religious activity and intrinsic religiousness. In the regression models total surgical fear was analyzed as a variable, not its aforementioned subscales.

To analyze potential confounders sociodemographic characteristics of the patients were assessed through a sociodemographic characteristics questionnaire with a focus on age, gender and educational status.

## Measures

A Croatian version of the Duke University Religion Index (DUREL) questionnaire was deployed. The Duke University Religion Index (DUREL), developed originally by Koenig et Büssing [34], is a short scale for the evaluation of basic religious or spiritual traits with good psychometric characteristics, which has already been translated into several languages and used in studies internationally/globally [18, 35–37]. It is a five-item measure of religious involvement, briefly assessing the three major dimensions of religiosity: organizational religious activity (ORA), non-organizational religious activity (NORA), and intrinsic (or subjective) religiousness (IR). The authors do not recommend summation into a total score [34] but literature has found DUREL to be a valid and reliable instrument for the assessment of religiousness in all the three dimensions. ORA and NORA are scored on a six-point Likert scale while the three IR items use a five-point Likert-type scale. While the lowest and highest scores on the subscales ORA and NORA are 1 and 6, the scores on subscale IR range from 3 to 15. Higher scores indicate a higher intensity of the specific dimension of religiousness. The Croatian version of the DUREL is a reliable instrument suitable to use, as validated by Murgić et al. [38].

The Croatian version of the Surgical Fear Questionnaire (SFQ) was used to determine the level of surgical fear [39]. The Surgical Fear Questionnaire developed by Theunissen et al. [15]. was chosen because of its high validity and reliability in all its international versions and its ability to differentiate between the fear of short-term and long-term consequences from

surgical procedures [1, 14, 16, 17]. The reliability and validity of the Croatian version have been demonstrated in a previous study by the authors [39]. The SFQ consists of eight items scored using a numeric rating scale. A score of 0 indicates "not afraid at all" and a score of 10 indicates "very afraid". The range of total scores lies between 0 and 80, with higher score indicating greater surgical fear. The Croatian version of the SFQ has been validated in an earlier study by the authors.

A sociodemographic questionnaire was developed for the purpose of this study. This form assessed 14 variables: Year of birth, gender, type of surgery, marital status (single, married, divorced, widowed), occupation, type of residence (flat, house, senior residence, rented flat, other) educational level (elementary, high school, intermediate, university), employment state (full-time, part-time, freelancing, other). Moreover, income level (less than 5000 Croatian Kuna, 5000–7500 Croatian Kuna, 7500–10000 Croatian Kuna, more than 10000 Croatian Kuna), smoking (number of cigarettes/time span in days, weeks, months), alcohol intake (never, occasionally, regularly, daily), and medication use were assessed.

The type of surgery was first verified in comparison with official patient documentation and then classified by the researchers upon data entry according to the following pattern. The indicated type of surgery was classified as either as a minor, intermediate or major elective surgical procedure. Minor surgical procedures were considered procedures from the proctologic domain and repairs of simple abdominal wall defects. Intermediate surgical procedures were considered all laparoscopic cholecystectomies and appendectomies and repair of complicated abdominal wall defects such as ventral postoperative or recurrent hernia. Major surgical procedures were considered all procedures targeting malignant disorders or bariatric procedures.

Additionally, subjects were asked whether they needed to use a sacral object (chapel, mosque, place of silence) the day before their procedure.

## Bias

The study protocol encountered two potential biases. One is sampling bias, since subjects may differ in some ways from the general population. The other is recall bias, which is present in all studies using self-reporting.

## Study size

The minimal sample size was based on the results of the study by Kalkhoran and Karimollahi [2], which included 150 participants. Sample size was then confirmed with power analysis. The Mann-Whitney U test, with significance level 0.05, including 178 participants, had at least 88% power to detect the medium effect size (d = 0.5). The Pearson's $X^2$ test had over 90% power to detect the medium effect size (w = 0.3), for 1, 2 and 3 difference contingency tables.

## Statistical analyses

For the analysis, all returned booklets were coded and data entry into the SPSS software (IBM Corp. Released 2020. IBM SPSS Statistics for Macintosh, Version 27.0. Armonk, NY: IBM Corp) was performed. Descriptive and inferential statistical analyses were conducted.

Scores on the religiousness and surgical fear scales and subscales were quantitative variables. After they were tested for the distribution type using the Kolmogorov-Smirnov test, the results of the descriptive analyses were presented as median and interquartile range. Differences in the distributions of the quantitative variables were analyzed with Mann-Whitney U tests.

The categorical independent variables were coded as follows: male and female gender as 1 and 2 respectively, the educational level from 1 to 4 with elementary and university education being 1 and 4 respectively and the type of surgery from 1 to 3 with minor and major surgical

procedures being 1 and 3 respectively. Their frequencies were presented as absolute and relative numbers. Differences in their distributions between groups were analysed with Pearson's $X^2$ test.

Predictive values of DUREL subscales on SFQ were analysed in series of univariate and multiple linear regression models. Residuals followed normal distribution. The regression coefficient (b) reflects the change in outcome (SFQ) for every unit of change of the respective independent variable.

### Ethical approval

The study protocol was approved by the Ethics Committee on Human Research of the University Hospital of Sveti Duh, Zagreb, Croatia confirmed by the approval letter 01-1095/15 on May the 28th, 2021. Signed informed consent was obtained from all participants.

### Results

The study was conducted between July 1st and September 30tht 2021. The non-probabilistic sample comprised 178 patients. The excess patient number can be attributed to effective patient recruitment by the researchers. Patient flow is shown below (Fig 1).

The study population was comprised of 106 male and 61 female subjects. Results are presented separately for men and women, and comparisons were conducted, given that previous

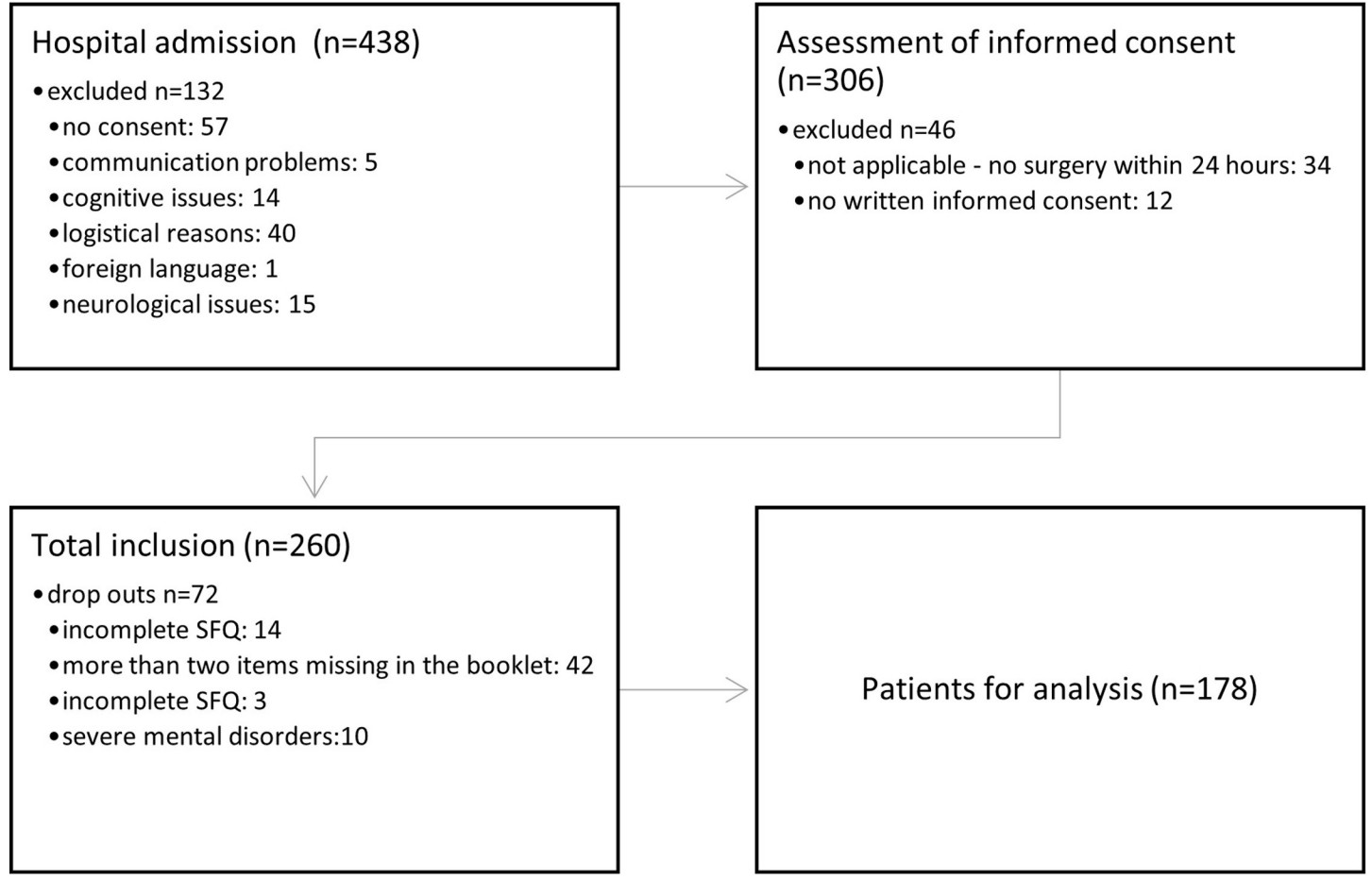

**Fig 1. Patient flow.**

research has identified gender to be a significant predictor of religiousness. All study results are publicly available in detail in the Supporting information section (S1 Table).

The results of descriptive statistical analyses for the quantitative variables are presented in Table 1.

The median age of the study population was 57 years, without statistically significant differences between men and women. In the context of religiousness men and women differed significantly regarding non-organizational and intrinsic religiousness, but not organizational religiousness. Female subjects indicated higher levels of non-organizational and intrinsic religiousness. Although the median score on the DUREL was higher in women, the difference was not statistically significant. Women reported significantly higher levels of surgical fear than men, the median in the female subgroup almost twice as big than in the male subgroup.

The descriptive analysis of the qualitative variables is disclosed in Table 2.

Men and women differed significantly regarding the type of surgery they underwent, but not regarding education level and the preoperative need for the use of a sacral object. The type of surgery was hence identified as an additional confounding factor, but not educational level and the preoperative need for the use of a sacral object.

The results of the univariate linear regression analysis evaluating relationships between surgical fear (SFQ) and religion (ORA, NORA, IR) are shown in Table 3.

All three measures of religiousness were statistically significant but weak predictors of surgical fear explaining 3% and 2% of variance respectively.

Multiple regression models were conducted to evaluate links between the dimensions of religiousness (the three DUREL subscales) and surgical fear, accounting for significant demographic and clinical covariates. Results of these analyses are presented in Tables 4 and 5.

All three dimensions of religiousness (ORA, NORA, IR) remained significantly correlated with surgical fear, controlling for potentially confounding factors. Notably, type of surgery was a strong predictor of surgical fear indicating that an increase in the complexity level of surgery explained greater surgical fear, by approximately 11 points on the SFQ scale. Together with age, gender, education and type of surgery, religious factors explained 20–21% of variance in surgical fear.

Subjects who expressed the preoperative need to use a sacral object reported higher levels of religiousness and surgical fear (for DUREL z = -5.73, p<0.001, for SFQ z = -2.01, p = 0.044).

**Table 1. Descriptive analysis of quantitative variables.**

|  | Men | Women | Total | Mann-Whitney U test | |
|---|---|---|---|---|---|
| Variable | Median (IQR) | Median (IQR) | Median (IQR) | Adjusted Z-value | P-value |
| Age | 57 (45–66) | 51 (41–65) | 57 (44–67) | 1.35 | 0.174 |
| *Religiousness* |  |  |  |  |  |
| DUREL | 12.5 (8–18) | 15 (10–20) | 14 (9–19) | 1.79 | 0.073 |
| ORA | 3 (2–3) | 3 (2–4) | 3 (2–3) | 0.22 | 0.828 |
| NORA | 1 (1–3) | 2 (1–4) | 1 (1–4) | 2.20 | 0.027 |
| IR | 9 (6–12) | 11 (7–12) | 9 (6–12) | 1.97 | 0.049 |
| *Surgical fear* |  |  |  |  |  |
| SFQ | 14.5 (6–27) | 28 (11–43) | 18 (8–34) | 3.67 | 0.001 |

IQR-interquartile range, DUREL-Duke Religion Index, ORA-Organizational religiousness, NORA-Non-organizational religiousness, IR-Intrinsic religiousness, SFQ-Surgical Fear Questionnaire

The total numbers for gender do not amount to the total number of study subjects due to missing data. 11 subjects did not indicate their gender.

**Table 2. Descriptive analysis of the qualitative variables.**

| Variable | Men (% of all men) | Women (% of all women) | Total (%) | Pearson Chi-square | | |
|---|---|---|---|---|---|---|
| | | | | Chi-square | Df | P-value |
| *Educational level* | | | | | | |
| Elementary education | 6 (5.83) | 5 (8.47) | 11 (6.79) | 2.158 | 3 | 0.532 |
| High school education | 64 (62.14) | 33 (55.93) | 97 (59.88) | | | |
| Intermediate education | 15 (14.56) | 13 (22.03) | 28 (17.28) | | | |
| University education | 18 (17.48) | 8 (13.56) | 26 (16.05) | | | |
| *Type of surgery* | | | | | | |
| Minor procedures | 57 (53.77) | 10 (22.95) | 67 (39.32) | 24.79 | 2 | <0.001 |
| Intermediate procedures | 26 (24.52) | 32 (52.46) | 58 (33.14) | | | |
| Major procedures | 3 (2.83) | 5 (8.19) | 8 (4.49) | | | |
| Preoperative use of a sacral object | 12 (11.32) | 7 (11.48) | 19 (11.38) | 0.001 | 1 | 0.975 |

The total numbers for gender, the education level and type of procedure do not amount to the total number of study subjects due to missing data. While 11 subjects did not indicate their gender, 16 subjects did not indicate their educational level, 45 subjects did not indicate the type of surgery they were undergoing.

## Discussion

Results from our study indicate positive relationships between the different dimensions of religiousness and surgical fear. This likely represents a mobilization effect, in which patients with higher levels of surgical fear activate religious beliefs and practices in response to their anxiety as a coping mechanism. This explanation is corroborated by our multivariate regression models, which found that patients who underwent more complex surgical procedures and reported higher scores regarding organizational religious activity, non-organizational religious activity or intrinsic religiousness respectively were at risk for higher levels of surgical fear, regardless of gender, age and education.

However, in statistical terms, religious factors alone explained around 2% of the variance in surgical fear. This is consistent with previous studies showing low levels of correlation between anxiety and religion, since positive and negative effects may cancel each other out in cross-sectional analyses.

The complexity in analyzing religiousness and the many ways its different dimensions may affect the patient's emotional and cognitive state [40, 41] is substantially due to an issue identified in a great body of scientific literature. Some research has questioned whether religiousness is a stable personal trait [42]or a dynamic, everchanging state, and this possibility is worthy of further investigation.

Literature on the relationship between religiousness and its facets and surgical fear is scarce. The only study which was primarily focused on the investigation of this association was by Kalkhoran and Karimollahi from 2007 [2]. This cross-sectional correlational study included 150 subjects and used the Spielberger trait anxiety inventory to assess preoperative anxiety and

**Table 3. Predictive values of DUREL subscales (ORA, NORA, IR) on surgical fear in univariate linear regression models.**

| Variable | F | Adj. $R^2$ | b regression coefficient | P-value | Intercept | P-value |
|---|---|---|---|---|---|---|
| ORA | 3.29 | 0.01 | 1.85 | 0.071 | 17.27 | <0.001 |
| NORA | 4.20 | 0.02 | 1.65 | 0.042 | 18.72 | <0.001 |
| IR | 6.20 | 0.03 | 0.86 | 0.014 | 14.71 | <0.001 |

ORA: organizational religious activity; NORA: non-organizational religious activity; IR: intrinsic religiousness

**Table 4. Predictive values of DUREL subscales (ORA, NORA, IR) on surgical adjusted for age, gender, education and surgery type in multiple linear regression models.**

| Variable | Model with ORA | | Model with NORA | | Model with IR | |
|---|---|---|---|---|---|---|
| | b regression coefficient | P-value | b regression coefficient | P-value | b regression coefficient | P- value |
| Intercept | -9.54 | 0.276 | -5.61 | 0.512 | -10.82 | 0.230 |
| Age | 0.11 | 0.271 | 0.13 | 0.188 | 0.13 | 0.196 |
| Gender | 3.71 | 0.234 | 2.39 | 0.456 | 3.21 | 0.310 |
| Education level | -0.65 | 0.701 | -0.93 | 0.584 | 0.10 | 0.955 |
| Type of surgery | 10.70 | <0.001 | 10.66 | <0.001 | 10.22 | <0.001 |
| ORA | 2.61 | 0.015 | - | - | - | - |
| NORA | - | - | 2.06 | 0.017 | - | - |
| IRA | - | - | - | - | 0.79 | 0.032 |

DUREL: Duke Religion Index; ORA: organizational religious activity; NORA: non-organizational religious activity; IR: intrinsic religiousness; Gender: 1 = male, 2 = female; Education level: elementary = 1, high school = 2, intermediate = 3, university = 4; Type of surgery: 1 = minor, 2 = intermediate, 3 = major

a questionnaire formulated by the researchers to assess religious belief. The study found a statistically not significant negative relationship between religiousness and preoperative anxiety ($r = -0.05$, $p = 0.49$) [2]. A cross-sectional study by Aliche at al. 2020 [31] investigated the effect of several psychological traits on preoperative anxiety on 210 subjects using the Spielberger trait anxiety inventory to assess preoperative anxiety and the Religious Commitment Inventory to assess religious commitment, which reflects the degree or level of religiousness. The study found interpersonal religious commitment, an analogue to religious activity to be negatively associated with preoperative anxiety [31]. A cross-sectional study from Muslu and Demir from 2020 [32], using the Anxiety specific to surgery Questionnaire and a questionnaire to assess religiosity formulated by the researchers, found that Muslim patients who perform religious rituals more often experienced lower preoperative anxiety levels before plastic surgery [32]. All these studies indicated religiousness, including different dimensions of religiousness, to be negatively related to preoperative anxiety.

Our study population reflected the high proportion of religious individuals in Croatia. According to the Croatian Population Census from 2011, 86.3% of the Croatian population declares themselves as Roman Catholics, while atheists, agnostics or skeptics solely amount to 5% [43]. This distribution is mirrored in our study population with 91.6% of subjects indicating at least low levels of religiousness while 8.4% not reporting any religiousness.

The transfer of our findings into clinical reality is limited due to the cross-sectional design of our study. Therefore, we cannot conclude causality in the association between the dimensions of religiousness and surgical fear. Religiousness and all its facets bear high levels of complexity which are difficult to investigate through a cross-sectional study.

Nevertheless, our study confirms one important hypothesis: religiousness is important in the perioperative setting and it is indeed in some way associated to surgical fear. Based on the

**Table 5. Summary of the multilinear regression models with the DUREL subscales.**

| Model | Adj. $R^2$ | F | P-value |
|---|---|---|---|
| Model with ORA | 0.21 | 7.76 | <0.001 |
| Model with NORA | 0.21 | 7.71 | <0.001 |
| Model with IR | 0.20 | 5.69 | <0.001 |

ORA: organizational religious activity; NORA: non-organizational religious activity; IR: intrinsic religiousness;

findings of our study findings several theoretical models could describe the statistically confirmed positive relationship between religiousness and surgical fear. It could either be that surgical fear incites religious activity or the reporting of the same. Or it could be that the reported religiousness is a context for the expression of surgical fear since people under stress do tend to think about high-order values which can engender struggles. Or finally it could simply be the result of religious individuals being at higher risk for surgical fear.

But nevertheless, two possible clinical implications of this study could be derived from this study. One is the potential beneficiary role of preoperative religion assessment in the evaluation and treatment of surgical fear [44]. A study by Taylor et al. found that 74% of surgical patients reported a possibly increased trust in their surgeon if they had inquired them about their religion preoperatively [45] indicating together with our study results that the preoperative assessment of religion is a valid component of the patient-surgeon relationship. The other is related to whether interventions based on religiousness represent a possible additional option for surgical fear management for religious patients [46, 47]. Several studies have shown that religion-based interventions such as prayers [48, 49], meditation [50] or religious counseling [51] and other religious practices [52] were helpful treatment options for patients dealing with anxiety [53, 54]. Our findings and current literature indicate that such measures could theoretically be of advantage in the preoperative setting for religious patients experiencing surgical fear. A possible religion-based intervention could be the recommendation or institutional organization of the use of sacral objects the day before surgery.

Despite an overwhelming religious majority of 92.9% and a fraction of 19.1% who pray at least once every day only 11.2% of study subjects reported the need to use a sacral object before undergoing surgery. The reason for such a small number is the object of speculation. The authors suspect that maybe hospitalized patients at our institution are not informed about the existence of easily accessible sacral objects in our facility. This matter needs to be clarified in future research more specifically.

Our results did show that patients who report the need for the use of sacral objects had higher levels of surgical fears and hence might be susceptible to such an intervention. Through the preoperative evaluation of patient´s religiousness by medical personnel one could select and guide patients to the use of sacral objects to reduce their surgical fear. Since no research has been conducted yet in scientific literature on religion-based interventions for surgical fear we recommend the investigation of this issue in the future. Considering the negative impact of surgical fear on the physical and psychological well-being of surgical patients, preoperative religion-based interventions could improve the overall success of surgical procedures. This study might help to lay foundations for the development of such interventions leading to enhanced preoperative treatment through the religious approach.

## Limitations

The main limitation of our study is the cross-sectional design of the study including only a 1-time measurement of surgical fear the day before surgery. Due to the plurality of mechanisms by which the different dimensions of religiousness might influence surgical fear, its effects cancel each out in data analysis yielding low effect sizes. Additionally, no causal relationships between religiousness and surgical fear can be inferred and the dynamics of surgical fear in the preoperative is insufficiently assessed.

Another limitation of the study is it being monocentric leading to significant sampling bias. Since all participants were recruited at single healthcare facility with patients gravitating from one region of Croatia (the capital and its surroundings), the representability of the results is limited. However, the sociodemographic composition of the study population did mirror the

general population. Due to great cultural differences pertaining religiosity, representative studies, preferably on national levels, need to be conducted in the future to provide adequate data on this subject. Focus needs to be on sociodemographic patients' traits which could act as confounders.

Another limitation further research is required to investigate the direction of the effect of religiousness and the possibility of a causal relationship between religiousness and surgical fear.

Another limitation is the fact that the religious identity of the subjects was not assessed due to the major prevalence of the Roman Catholic faith. Possibly, different religious affiliations might affect surgical fear in different ways.

## Conclusion

This study showed a small but statistically significant positive association between the dimensions of religiousness and surgical fear, potentially suggesting that individuals who are more afraid of surgery the day before the procedure tend to harness religion as a coping resource. Alternatively, it is possible that religious individuals experience greater surgical fear. While more research is required to address directions of effect, our results certainly highlight the importance of religiousness assessment and religion-based interventions in the context of surgical fear.

## Supporting information

**S1 Table. Study results.**
(XLSX)

## Acknowledgments

We thank all the participants who contributed to this research and prof. Žarko Rašić, the head of the Surgery Department at University Hospital Sveti Duh.

## Author Contributions

**Conceptualization:** Andrija Karačić.

**Data curation:** Andrija Karačić, Jure Brkić.

**Formal analysis:** Slavica Sović, Jelena Karačić.

**Investigation:** Branko Bakula.

**Supervision:** Andrija Karačić.

**Validation:** Andrija Karačić.

**Writing – original draft:** Andrija Karačić.

**Writing – review & editing:** Maurice Theunissen, Mansoureh Karimollahi,
David H. Rosmarin.

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
