## [Decision Letter · Decision Letter 0]

4 Apr 2023

PONE-D-22-33682Are religious patients less afraid of surgery? A cross-sectional study on religiousness and surgical fearPLOS ONE

Dear Dr. Andrija Karačić

Thank you for submitting your manuscript to PLOS ONE. After careful consideration, we feel that it has merit but does not fully meet PLOS ONE’s publication criteria as it currently stands. Therefore, we invite you to submit a revised version of the manuscript that addresses the points raised during the review process.

Please answer carefully revisor's questions and correct the manuscript according to their suggestions.

We look forward to receiving your revised manuscript.

Kind regards,

Kornelia Zaręba, MD

Academic Editor

PLOS ONE

Journal Requirements:

Additional Editor Comments:

Reviewers' comments:

Reviewer's Responses to Questions

**Comments to the Author**

1. Is the manuscript technically sound, and do the data support the conclusions?

Reviewer #1: Yes

Reviewer #2: Partly

Reviewer #3: Yes

2. Has the statistical analysis been performed appropriately and rigorously? 

Reviewer #1: Yes

Reviewer #2: Yes

Reviewer #3: Yes

3. Have the authors made all data underlying the findings in their manuscript fully available?

Reviewer #1: Yes

Reviewer #2: Yes

Reviewer #3: Yes

4. Is the manuscript presented in an intelligible fashion and written in standard English?

Reviewer #1: Yes

Reviewer #2: Yes

Reviewer #3: Yes

5. Review Comments to the Author

Reviewer #1: Title suggest to change

Relationship between preoperative surgical fear and religiosity dimensions

Line 53 cross sectional study among surgical clinics in one hospital?

Line 54 setting identify the name of the tertiary facility /hospital

Line 78 key words :- no need to repeat preoperative

94 could you mention the study significance prevalence of fear among Croatian patients

Line 116 needs to be dissected, taking into account the different potential effects ( need rephrases )

161 elective surgery you mean ( regardless the type of surgery ) ?

Reviewer #2: I appreciate the opportunity given to review this interesting manuscript on religiousness and surgical fear.

The focus of the study is clearly defined in terms of what aspects of religiousness are under observation (organized religiousness, non-organized religiousness, and intrinsic religiousness). The originality of the research is unquestionable as it differs from other literature on the link between surgical fear or preoperative anxiety and religiousness. The study also shows adequate knowledge and understanding of relevant literature on the concept.

The variables used in the study are well-defined.

The discussion section is well written. The authors make adequate references to other studies that relate to the outcome of their study. Also, the sentences are clearly expressed and readable.

However, in lines 197 and 200, the authors did not cite any literature to support their statements.

Also, the authors did not provide any form of interpretation or discussion for the results presented in Table 1 and did not provide enough discussion for the results in Table 2.

Again, tables in the results section are not presented in APA format which makes interpreting results from the tables quite challenging. Especially for Table 4.

In addition, there are a few grammatical errors identified. For instance, the authors wrote, “further” with a capital F in line 392. It is recommended that the authors would revise the manuscript to check for grammatical errors and well as present tables in APA format with an adequate interpretation of findings.

Again I am grateful for the opportunity to review this manuscript and hope to see it in print.

Reviewer #3: Dear authurs, I have reviewied your manuscript. I found it very interesting. The topic, procedures and methodology the langugue .data analises. discusion and conclusion too. I also apreciate and value the role of religiousness on different areas. In short Ido have minor and a few coments to be cheched

1. Something missed between the subtopic of variables and stastical analysis on page 6…

2. 8.4% of your respondents not reported any religious responses.page 10. probably good to compare these respondents with others,.because you may find valuabel information.

3. If you check your refrences from the perspective of APA Formating style, eg journal names need to be italic

6. PLOS authors have the option to publish the peer review history of their article (what does this mean?). If published, this will include your full peer review and any attached files.

Reviewer #1: No

Reviewer #2: No

Reviewer #3: **Yes: **Dereje Adefirs

---

## [Author Response · Author response to Decision Letter 0]

19 Apr 2023

Response to Reviewers

Dear Reviewers,

thank you very much for your constructive review and all the compliments. I hope to have incorporated all your commentaries into the new version of the manuscript.

Below my replies to the reviewers:

- as suggested, I have changed the manuscript title by adding the aspect of dimensions.

- I have specified the details in the Design and Setting subsection of the Abstract as instructed. 

- I have crossed out the repeated term in the Key words.

- I cannot mention the prevalence of fear among Croatian patients, because up-to-date only one study has been completed by our group. But, this study did not assess the presence or absence of surgical fear but the level of surgical fear, so we cannot present data on how many patients in Croatia indicate surgical fear, respectively indicate no surgical fear at all. 

- I have rephrased the sentence in Line 116, now 117 and hope that this formulation is more comprehensible.

- in line 161 I have written that all patients admitted for any elective surgery, regardless of type of surgery were potential candidates. 

- I have added the reference to our previous study on surgical fear where we have validated the Croatian version of the SFQ, which has been accepted for publication, but has not yet been published.

- I have added passages in the Results section where I have commented on the results of Table 1 and extended my commentaries on Table 2. 

- Table 4 was broken up into two tables to render a clearer graphic representation of the respective results. 

- A thorough check for grammatical error in the whole manuscript, not only line 392 was performed.

- The suspected missing part in the Methods section was a place-holder and deleted.

- The reviewers suggested to disclose more information on the patient population who reported no religious behaviors. This was a thought of ours, but since the manuscript focuses on religious patients and their relationship with surgical fear, we did not include data on this specific topic. Additionally, we realized that data analysis on this subject would yield only information about sociodemographic factors regarding this subgroup, not novel insight into the relationship between religiousness and surgical fear. 

- All references have been checked for the APA style. 

Kind regards,

Andrija

---

## [Decision Letter · Decision Letter 1]

6 Jun 2023

Are religious patients less afraid of surgery? A cross-sectional study on the relationship between dimensions of religiousness and surgical fear

PONE-D-22-33682R1

Dear Dr. Andrija Karačić,

We’re pleased to inform you that your manuscript has been judged scientifically suitable for publication and will be formally accepted for publication once it meets all outstanding technical requirements.

Kind regards,

Othman A. Alfuqaha, Ph.D.

Academic Editor

PLOS ONE

Additional Editor Comments (optional):

Congratulations for your work. However, the authors could revise the statement in line 416. Also, reference 13 should be revised to suit the plos one journal. Please, double check to all references.

Reviewers' comments:

Reviewer's Responses to Questions

**Comments to the Author**

1. If the authors have adequately addressed your comments raised in a previous round of review and you feel that this manuscript is now acceptable for publication, you may indicate that here to bypass the “Comments to the Author” section, enter your conflict of interest statement in the “Confidential to Editor” section, and submit your "Accept" recommendation.

Reviewer #1: All comments have been addressed

Reviewer #2: All comments have been addressed

Reviewer #3: All comments have been addressed

2. Is the manuscript technically sound, and do the data support the conclusions?

Reviewer #1: Yes

Reviewer #2: Yes

Reviewer #3: Yes

3. Has the statistical analysis been performed appropriately and rigorously? 

Reviewer #1: Yes

Reviewer #2: Yes

Reviewer #3: Yes

4. Have the authors made all data underlying the findings in their manuscript fully available?

Reviewer #1: Yes

Reviewer #2: Yes

Reviewer #3: Yes

5. Is the manuscript presented in an intelligible fashion and written in standard English?

Reviewer #1: Yes

Reviewer #2: Yes

Reviewer #3: Yes

6. Review Comments to the Author

Reviewer #1: the authors have changed the comments, provided to them , I think the article is proofed for publication.

Reviewer #2: Comments made earlier have been addressed by the authors.

However, the authors could revise the statement in line 416. Also, reference 13 should be revised to suit the APA style.

Thank you.

Reviewer #3: Dear authors I have found this article more interesting. Besides, I want to thank you for addressing all comments and suggestions in professional ways.

7. PLOS authors have the option to publish the peer review history of their article (what does this mean?). If published, this will include your full peer review and any attached files.

Reviewer #1: No

Reviewer #2: No

Reviewer #3: No

---

## [Editor Report · Acceptance letter]

4 Jul 2023

PONE-D-22-33682R1 

Are religious patients less afraid of surgery? A cross-sectional study on the relationship between dimensions of religiousness and surgical fear 

Dear Dr. Karačić:

I'm pleased to inform you that your manuscript has been deemed suitable for publication in PLOS ONE. Congratulations! Your manuscript is now with our production department. 

Kind regards, 

on behalf of

Dr. Othman A. Alfuqaha 

Academic Editor

PLOS ONE